# The Significance of mRNA in the Biology of Multiple Myeloma and Its Clinical Implications

**DOI:** 10.3390/ijms222112070

**Published:** 2021-11-08

**Authors:** Anna Puła, Paweł Robak, Damian Mikulski, Tadeusz Robak

**Affiliations:** 1Department of Hematology, Medical University of Lodz, 93-510 Lodz, Poland; anna.pula@stud.umed.lodz.pl; 2Department of Experimental Hematology, Medical University of Lodz, 93-510 Lodz, Poland; pawel.robak@umed.lodz.pl; 3Department of Biostatistics and Translational Medicine, Medical University of Lodz, 92-215 Lodz, Poland; damian.mikulski@stud.umed.lodz.pl

**Keywords:** biology, drug resistance, gene expression profiling, mRNA, multiple myeloma, prognosis, treatment

## Abstract

Multiple myeloma (MM) is a genetically complex disease that results from a multistep transformation of normal to malignant plasma cells in the bone marrow. However, the molecular mechanisms responsible for the initiation and heterogeneous evolution of MM remain largely unknown. A fundamental step needed to understand the oncogenesis of MM and its response to therapy is the identification of driver mutations. The introduction of gene expression profiling (GEP) in MM is an important step in elucidating the molecular heterogeneity of MM and its clinical relevance. Since some mutations in myeloma occur in non-coding regions, studies based on the analysis of mRNA provide more comprehensive information on the oncogenic pathways and mechanisms relevant to MM biology. In this review, we discuss the role of gene expression profiling in understanding the biology of multiple myeloma together with the clinical manifestation of the disease, as well as its impact on treatment decisions and future directions.

## 1. Introduction

Multiple myeloma (MM) is a genetically complex disease resulting from a multistep transformation of normal to malignant plasma cells in the bone marrow [1]. Its precursors are believed to be monoclonal gammopathy of undetermined significance (MGUS) and smoldering multiple myeloma. However, while both lack the clinical features of organ damage presence, such as hypercalcemia, renal insufficiency, anemia, and bone lesions, they share some genetic mutations of symptomatic MM [2,3]. Further progression of the disease may lead to the proliferation of clonal plasma cells at sites outside the bone marrow, manifesting as extramedullary myeloma and plasma cell leukemia (PCL), both known to be very aggressive malignancies with inferior outcomes [4].

As MM occurs mainly in older patients, its treatment has gained prominence in today’s aging population. Its annual incidence in the United States in 2020 was estimated to be as high as 4–6 cases per 100,000, with 32,270 new cases and 12,830 deaths reported [4,5].

In the era of molecular cytogenetic methodologies such as G-band karyotyping, fluorescence in situ hybridization (FISH), comparative genomic hybridization (CGH), as well as more advanced novel genetic techniques, such as single nucleotide polymorphism (SNP) arrays and next-generation sequencing (NGS), it has become possible to better understand the molecular background of myelomagenesis [6]. Multiple myeloma is a genetically heterogeneous disease. The genetic alterations present in MM can be categorized into translocations, copy number abnormalities (CNAs), and point mutations [7,8]. The most important molecular mechanism underlying MM pathogenesis is thought to be immunoglobulin heavy chain (IgH) translocation [9]. Although the molecular mechanisms responsible for the initiation and heterogeneous evolution of MM remain largely unknown to date, the identification of driver mutations is fundamental to understanding the oncogenesis of MM and its response to therapy. However, the genetic landscape of MM is very complex, and distinguishing driver from passenger mutations is challenging. The somatic mutation rate of patients with multiple myeloma was reported to be approximately 1.6 mutations per Mb [10]. Certain genes, including *KRAS, NRAS, TP53, FAM46C, DIS3*, and *BRAF* have been reported to demonstrate frequent mutations in myeloma patients [11,12,13].

The introduction of gene expression profiling (GEP) in MM was an important step in elucidating the molecular heterogeneity of MM and its clinical relevance. Initially array-based studies, and more recently, those based on RNA sequencing (RNASeq), provided information on the transcriptomic background of myeloma, its clinical course, and prognosis. Since some mutations in MM occur in non-coding regions [14], analytical approaches based on mRNA provide more comprehensive information on the oncogenic pathways and mechanisms relevant to MM biology.

This present review discusses the role of gene expression profiling in understanding the biology of MM, together with the clinical manifestation of the disease, as well as its impact on treatment decisions and future directions for research.

## 2. Techniques Used for Gene Expression Analysis

The history of transcript profiling begins with early attempts of Northern blotting, reverse transcriptase quantitative PCR (RT-qPCR), and Sanger sequencing of the expressed sequence tags (ESTs), these being short nucleotide sequences generated from cDNAs [15,16,17,18]. Other early gene expression analysis techniques include serial analysis of gene expression (SAGE) [19] and DNA microarrays [20]. Both techniques are widely used for gene expression studies and novel gene identification. SAGE is based on the principle that an oligonucleotide sequence can uniquely identify a gene. It requires the isolation of mRNA and the generation of cDNA, from which unique small sequences (∼initially 10 bp), i.e., tags, are generated using restriction enzyme digestion. The frequency of a specific sequence tag determines the relative abundance of the transcript. Over the years, variations of SAGE have been devised to identify tags more accurately by increasing tag length by even as much as 26 bp [21]. DNA microarrays act by measuring the hybridization of the labeled target cDNA strands to a sample with fixed probes [22]. Although the techniques mentioned have been widely used, they both have their limitations. 

The development of the high throughput sequencing RNA-seq technique has enabled even better exploration of RNA biology. The popularity of RNA-seq is driven by its large number of applications with differential gene expression analysis being the most common one. The standard workflow of RNA-seq begins with RNA extraction. This is followed by the purification of RNA from a sample since the isolated RNA is mostly ribosomal. The two most common techniques used for target enrichment are poly(A) capture for mRNA selection and ribosomal depletion. Following this, cDNA synthesis is performed and an adaptor-ligated sequencing library is prepared. Finally, the cDNA library is amplified by polymerase chain reaction (PCR) using parts of the adapter sequences as primers. When the experiment is finished, the data analysis begins: aligning and/or assembling the sequencing reads to a transcriptome, quantifying reads that overlap transcripts, filtering and normalizing between samples, and statistical modeling of significant changes in the expression levels of individual genes and/or transcripts between sample groups [23,24].

Over the years, our understanding of hematological malignancies has improved thanks to the development of next-generation sequencing (NGS), an approach comprising a range of methodologies that allow the investigation of genomics, transcriptomics, and epigenomics. An extensive review by Braggio et al. details the advances in the genomic exploration of hematological malignancies achieved through genome-wide sequence analysis [25].

Transcriptomic studies have provided important information regarding pathways and genes involved in myelomagenesis. Such gene expression profile (GEP) studies constitute a reliable prognostic tool that has been independently validated by various multiple myeloma cooperative groups. However, in daily clinical practice, no consensus has evolved to integrate GEP in multiple myeloma care.

## 3. Gene Expression Profile in Multiple Myeloma Biology and Prognosis

Multiple myeloma is a genetically complex and heterogeneous neoplasm in which the concurrency of multiple genomic events results in tumor development and progression. MM exists as hyperdiploid and nonhyperdiploid forms, with different karyotype [26,27]. Its most important oncogenic mechanisms are believed to be oncogene activation by IgH translocations and oncogene mutations [28]. IgH translocations are present in up to 50% of patients, and mainly involve five chromosomal loci, 11q13, 6p21, 4p16, 16q23, and 20q11, which contain the CCND1, CCND3, *FGFR3/NSD2, MAF*, and *MAFB* oncogenes [29].

The transcriptome of multiple myeloma has been evaluated in different patient cohorts [30,31,32,33]. Studies based on GEP have been widely used to better understand the biology of MM by identifying the genes involved in the molecular pathogenesis of the disease and their clinical significance, to predict survival in multiple myeloma, and to identify patients who will benefit from particular types of therapy. Some groups have even made an attempt to compare the transcriptome of MM and primary plasma cell leukemia: a more aggressive form of plasma cell dyscrasia [34]. Expression profiles of differentially expressed genes are of critical importance and have provided insights into MM biology. These genes may relate to cell cycle, cell death, autophagy, kinome, stemness, cytogenetic abnormalities, chromosome 1, homozygous deletions, and immune subnetworks [33,35,36,37,38,39,40,41,42].

GEP studies have led to the identification of Cyclin D family deregulation in MM and MGUS [30,43,44]. Deregulation of the cyclin D family (CCND1, CCND2, and CCND3) appears to be one of the key molecular events in the pathogenesis of MM [45]. It can result from the translocation of CCND1 or CCND3 with the *IgH* gene in the t(11;14) and the t(6;14), specific cyclin D amplification, trisomies, and other cytogenetic events. CCND2 is particularly overexpressed in t(4;14) and t(14;16) patients [30,31]. A proposed classification based on CCND1 gene expression status and 14q32 translocations divides MM patients into eight different subgroups [44]. 

Another attempt to use gene expression profiling in order to develop a prognostically relevant molecular classification of MM was made by Zhan et al. [32] The findings indicated the presence of seven disease subtypes that were strongly influenced by known genetic lesions including *c-MAF–* and *MAFB-*, CCND1- and CCND3-, *MMSET*-activating translocations and hyperdiploidy, these being CD1 [(t(11;14)], CD2 [t(11;14) and t(11;16)], MS [t(4;14)], MF [t(14;16) and t(14;20)], hyperdiploid cluster (HY), low bone disease (LB), and proliferation-associated genes (PR). Zhan et al. also identified myeloid gene expression signatures but excluded them from profiling analyses [32]. Broyl et al. confirmed the findings made by Zhan et al. and identified three novel subsets of MM: the nuclear factor kappa light chain-enhancer (NF-kB) subgroup, the cancer/testis antigen (CTA) subgroup characterized by high proliferation index, and the PRL3 subgroup characterized by up-regulation of protein tyrosine phosphatases PRL-3 and PTPRZ1 [31].

A review by Szalat et al. indicated the existence of 11 different molecular subgroups of MM based on transcriptomic studies [46]. A summary of this classification correlated with the clinical outcome is given in Table 1. Liu et al. combined data from whole-genome gene expression profiling microarrays and CytoScan HD high-resolution genomic arrays to integrate GEP with copy number variations (CNV); the findings highlighted certain molecular alterations in MM that were important for disease initiation, progression, and poor clinical outcome. In particular, eight cytogenetic driver lesions essential to the development and progression of myeloma were highlighted by the amplification of chromosome 1q: they suggest that 1q gains and the upregulated *ANP32E, DTL, IFI16, UBE2Q1,* and *UBE2T* gene expression could be responsible for MM aggressiveness [47]. These findings support those of Shaughnessy et al., who found that most of the up-regulated genes mapped to chromosome 1q, and the down-regulated genes mapped to chromosome 1p; this suggests that disease progression may be influenced by changes in the transcriptional regulation of genes mapping to chromosome 1 [40]. However, studies based on different molecular methods have yielded conflicting findings considering 1q gain as an adverse prognostic factor. Some early studies suggest it has no prognostic value [48,49], while some latest reports suggest it may be associated with an inferior outcome [50,51,52,53].

Manasach et al. compared the value of retrospective GEP data with FISH criteria to identify high-risk (HR) patients. They conclude that GEP identified more HR patients than FISH. Patients reclassified from standard-risk FISH to HR GEP presented with 1q amplification of equal to or over four copies [54]. Elsewhere, a multi-tissue transcriptome-wide association study (TWAS) aimed at exploring MM biology by Went et al. [55] identified 108 genes at 13 independent regions associated with MM risk; all of these were within 1 Mb of known MM GWAS risk variants [56,57,58,59].

It should be noted that transcriptomic approaches have rarely been employed in assessments of the risk of multiple myeloma or progression from MGUS. A number of GWAS and SNP studies have been conducted in order to explore this field, including multiple studies by the International Multiple Myeloma Research (IMMEnSE) consortium [56,57,58,59,60,61,62].

## 4. Gene Expression Profile and Multiple Myeloma Prognosis

Many different transcriptomic models for prognostication have been identified; however, none of them have been introduced into routine clinical practice. So far, the revised International Staging System (R-ISS) is still the first choice in MM management [63], and the older Durie-Salmon staging system is still used in some places [64]. Zhan et al. performed a microarray analysis on tumor cells from 532 newly diagnosed patients with MM in order to identify high-risk disease [32]. They report that high-risk groups presented a similar gene expression profile to human MM cell lines, whereas low-risk MM groups exhibited patterns identical to MGUS and normal plasma cells. After evaluation of the 70-gene risk model in relapse samples of 51 out of 351 of the training cohort, high-risk scores associated with poor survival were found in 39 patients. Kuiper et al. identified a 92-gene signature (EMC-92) that proved to be an independent prognostic factor of survival [65]. More recently Decaux et al. proposed a risk stratification model based on 15 different genes and note that patients with high-risk MM were characterized by the overexpression of genes involved in multiple phases of the entire cell cycle [33]. Dickens et al. limited the prognostication to six genes [41]. Similarly, Botta et al. proposed a prognostic risk score based on only six genes: *IFNG, IL2, LTA, CCL2, VEGFA,* and *CCL3* [42]. This list was acquired from a gene expression profiling dataset of MGUS, smoldering MM, and symptomatic-MM, and identified inflammatory and cytokine/chemokine pathways as the most progressively affected during disease evolution.

Hose et al. proposed that assessment of proliferation by GEP allows the selection of patients for risk-adapted anti-proliferative treatment [66]. Liu et al. [35] constructed a multiple myeloma molecular causal network (M3CN) based on gene expression, copy number variation, and clinical data to better understand MM tumorigenesis, progression, and drug responses. The M3CN-derived prognostic subnetwork achieved demonstrated satisfactory separation between different risk groups [35]. However, the most complex approach was proposed by Katiyar et al. [67], who identified unified potential signatures for MM based on a genome-wide meta-analysis of differentially expressed genes (DEGs) and miRNAs (DEMs) in MM cells and normal plasma cells. The authors identified the top five most functionally connected hub genes *(UBC, ITGA4, HSP90AB1, VCAM1, VCP*) using protein–protein interactions. 

In addition, transcription factor regulatory networks were determined for five seed DEGs with four or more biomarker applications (*CDKN1A, CDKN2A, MMP9, IGF1, MKI67*) [67]. The above studies indicate, that DEGs may influence disease pathogenesis, clinical presentation, and drug sensitivities in MM patients. 

In recent years, gene expression profiling has been used to establish classifiers for prognostication. Various studies have shown that that GEP classifiers are more robust than FISH markers in identifying risk. For instance, a multivariate analysis by Kuiper et al. found that combinations of GEP with ISS, particularly SKY92 + ISS, proved superior to other combinations for stratifying MM into high-risk and low-risk categories [68]. A summary of the differences between gene expression classifiers in MM is presented in Table 2.

## 5. mRNA and Drug Resistance

Despite recent advancements in the design of novel anti-myeloma drugs, the acquisition of anti-cancer drug resistance is a major limitation of MM therapy. The mechanisms underlying drug resistance are diverse and include both genetic and epigenetic abnormalities. The topic of drug resistance in multiple myeloma has been widely reviewed by Robak et al. [73]. However, for the purpose of this review, we would like to briefly mention the mechanisms associated with altered mRNA expression.

Mitra et al. [74] developed a gene expression signature that predicts response specific to proteasome inhibitor (PI) treatment in MM on human myeloma cell lines (HMCLs). They created a 42-gene expression signature that could distinguish good and poor PI response in the HMCL panel and could be successfully applied to four different clinical data sets on MM patients undergoing PI-based chemotherapy to distinguish between good and poor outcomes [74].

In a study of the functional role of ABCB1 overexpression in MM, Besse et al. [75] found this to be the most significant change in carfilzomib-resistant MM cells compared to bortezomib-resistant cells. This change enhances the *p*-glycoprotein-mediated export of therapeutic drugs. The authors identified nelfinavir and lopinavir as approved drugs that could overcome resistance to carfilzomib by modulating P-glycoprotein function [75]. In addition, they observed that *ABCB1* overexpression reduces the proteasome-inhibiting activity of carfilzomib but not of bortezomib.

Tang et al. [67]. identified 2099 long non-coding mRNAs that were deregulated in exosomes of bortezomib-resistant patients. Of these, 78 mRNAs in drug resistance-related pathways were enriched, with mammalian targets: rapamycin, platinum drug resistance, the cAMP, and phosphoinositide 3-kinase/Akt signaling pathways being key examples [76].

A recent study by Robak et al. [77] compared the mRNA expression of nine previously described genes that may affect resistance to multiple myeloma (*ABCB1, CXCR4, MAF, MARCKS, POMP, PSMB5, RPL5, TXN*, and *XBP1*) by bortezomib-refractory and bortezomib-sensitive patients [77]. The analysis was performed on 73 MM patients and 11 healthy controls. It was reported that *RPL5* was significantly downregulated in MM patients, and that POMP was significantly upregulated in MM patients refractory to bortezomib. A multivariate analysis found high expression of *PSMB5* and *CXCR* and autologous stem cell transplantation to be independent predictors of progression-free survival, while high expression of *POMP and RPL5* was associated with shorter overall survival [77].

## 6. mRNA in CAR-T Cell Therapy

When reviewing the role of messenger RNA in the biology of multiple myeloma, it is important to include the latest achievement in the field of chimeric antigen receptor (CAR) T cell therapy. Despite the introduction of many novel therapeutic strategies, multiple myeloma remains incurable and requires continued intervention for disease control. However, a promising recent development is the design of an engineered T-cell product, Descartes-08, that transiently modifies a purified population of autologous CD8 + T-cells with anti-B cell maturation antigen (BCMA) CAR mRNA, as reported by Lin et al. [78]. Descartes-08 is engineered by mRNA transfection to express anti-BCMA CAR for a defined length of time. The mRNA is synthesized by in vitro transcription from a linearized DNA plasmid [78]. The development of this virus-free CAR-T cell technology has recently led to the initiation of the first clinical trial [79].

## 7. Conclusions

Gene expression profiling studies provide important information regarding the biology of multiple myeloma and may serve as a tool to predict outcomes and guide therapy. In the era of personalized medicine, the future lies in enabling therapy to be chosen based on the presence of specific mutations and gene expression profiles. However, the complexity of the MM genome and transcriptome still requires further investigation.

## Figures and Tables

**Table 1 ijms-22-12070-t001:** The identification of 11 molecular subgroups of incorrectly expressed genes using gene expression profiling.

Prognosis	Subgroup	Cytogenetics	Cyclin D Expression	Upregulated Genes	Downregulated Genes	Frequency
Low risk	CD1	t(11;14)	CCND1	*INHBE* *ETV1* *MACROD2*	*CD9* *NOTCH2NL*	4–9%
	CD2	t(11;14) t(6;14)	CCND1 CCND3	*cd79a* *cd20*	CCND2	11–17%
	LB	-	CCND1 CCND2	*EDN1* *IL6R* *SMAD1*	*DKK1* *STAT1* *STAT2*	12–17%
	HY	HD	CCND1	*TRAIL* *DKK1* *CCR5*	CCND2 *CD52* *TAGLN2* *CKS1B* *OPN3*	26–32%
	NF-κB	HD	CCND1 CCND2	*CD40* *BCL10* *IL8*	*TRAF3* *CCR2* *MAT2A*	11%
	PRL3	HD	CCND2	*SOX3* *PTP4A3* *PTPRZ1*	*CD44* *DUSP6*	2–3%
	Myeloid	-	CCND1 CCND2	*CD163* *CA1* *LIZ*	*PRMT1* *DUSP5* *SMAD7*	12%
High risk	MF	t(14;16) t(14;20)	CCND2	*IL6R* *c-MAF* *MAFB*	*DKK1* CCND1	6–10%
	MS	t(4;14) 1q gain	CCND2	*MMSET* *FGFR3* *PBX1*	CCND1 *DUSP2* *SYK PAX5*	15–17%
	PR	1q gain	CCND2 CCND1	*CCNB1* *MCM2* *CDC2* *BIRC5* *CCNB2* *AURKA*	*CXCR4* *CD27*	11%
	CTA	1q gain	CCND1 CCND2	Cancer testis antigen *AURKA*	*MALAT1*	7%

**Table 2 ijms-22-12070-t002:** Summary of differences between gene expression classifiers in MM.

Classifiers Name	No. of Genes Involved	Gene Names Involved	No. of Patients Tested in Validation in the Original Publication	No. of Risk-Groups Identified	Reported Outcome	Ref.
UAMS70	70	*FABP5, PDHA1, TRIP13, AIM2, SELI, SLCI19A1, LARS2, OPN3, ASPM, CCT2, UBE2I, STK6, FLJ13052, LAS1L, BIRC5, RFC4, CKS1B, CKAP1, MGC57827, DKFZp779O175, PFN1, ILF3, IFI16, TBRG4, PAPD1, EIF2C2, MGC4308, ENO1, DSG2, C6orf173, EXOSC4, TAGLN2, RUVBL1, ALDOA, CPSF3, NA, MGC15606, LGALS1, RAD18, SNX5, PSMD4, RAN, KIF14, CBX3, TMPO, DKFZP586L0724, WEE1, ROBO1, TCOF1, YWHAZ, MPHOSPH1, GNG10, NA, PNPLA4, NA, KIAA1754, AHCYL1, MCLC, EVI5, AD-020, NA, PARG1, CTBS, UBE2R2, FUCA1, RFP2, FLJ20489, NA, LTBP1, TRIM33*	412	7	Low-risk groups 3-year EFS: 84% in LB, 72% HY, 82% in CD1, and 86% in CD2 3-year OS: 81% in CD1, 84% in HY, 87% in LB, and 88% in CD2	[32]
					High risk groups 3-year EFS: 44% in PR, 39% in MS and 50% in MF 3-year OS: 55% in PR, 69% in MS, 71% in MF	
UAMS17	17	*KIF14, SLC19A1, CKS1B, YWHAZ, MPHOSPH1, TMPO, NADK, LARS2, TBRG4, AIM2, 242488_at, ASPM, AHCYL1, CTBS, MCLC, LTBP1, 1557277_a_at*	181	2	High-risk vs. low-risk 5-year EFS: 18% vs. 60%, *p* < 0.001; HR = 4.51 5-year OS: 28% vs. 78%, *p* < 0.001; HR = 5.16	[40]
UAMS80	80	*COX6C, NOLA1, COPS5, SOD1, TUBA6, HNRPC, PSMB2, PSMC4, LOC400657, C1orf31, FUNDC1, SUMO1, PSMB4, PSMB3, ENSA, PSMB4, COMMD8, MRPL47, PSMC5, HNRPC, PSMA4, PSMD4, NMT1, PSMB7, NXT2, SLC25A14, PSMD4, PSMD2, SNRPD1, PSMD4, CHORDC1, PSMD14, LAP3, PSMA7, UBPH, BIRC5, STAU2, ALDOA, TMC8, C1orf128, FLNA, HIST1H3B, FOSB, LOC644250, C17orf60, LZTR2, PDE4B, STAU2, PDE4B, GABARAPL1, TAGAP, LOC643318, CISH, NR4A1, MGC61598, ANKRD37, KIAA1394, ACVR1C, TBC1D9, CRYGS, PDE4B, CISH, ZNF710, RBM33, STX11, KIAA1754, CISH, RPL41, WIRE, LAPTM4A, KLHL7, C9orf130, C14orf100, 1561060_at, 229388_at, 239343_at, 236986_at, 227524_at, 239082_at, 226399_at*	128	2	Low-risk vs. high-risk 2-year OS: 92% vs. 60% 2-year PFS: 87% vs. 53%	[69]
IFM15	15	*CNDP2, STMN1, AFG3L2, STK38, PARP1, CPSF6, LOC151162, TOX2, FRY, FLJ21438, MGST1, ALDH2, CTSF, ATF4, FAM49A*	853	2	3-year OS Low-risk: 90.5% (95% CI, 85.6% to 95.3%) High-risk: 47.4% (95% CI, 33.5% 60.1%)	[33]
MRCIX6	6	*BUB1B, HDAC3, CDC2, FIS1, RAD21, ITM2B*	800	2	Median OS: 13 vs. 45 months Median PFS 11 vs. 22 months	[41]
HM19	19	*BUB1B, HEATR2, CENPW, NUDT11, EHD4, EZH2, DLGAP5, HJURP, CDCA8, TMEM48, CDC42BPA, DEPDC1B, FAM83D, PGM2L1, NUSAP1*	345	3	OS at 48 months (%): 92 vs. 72 vs. 20	[70]
GPI50	50	*ASPM, AURKA, AURKB, BIRC5, BRCA1, BUB1, BUB1B, CCNA2, CCNB1, CCNB2, CDC2, CDC20, CDC25C, CDC6, CDCA8, CDKN3, CEP55 (C10orf3), CHEK1, CKS1B, CKS2, DLG7, ESPL1, GINS1, GTSE1, KIAA1794, KIF11, KIF15, KIF20A, KIF2C, KNTC2, MAD2L1, MCM10, MCM6, MKI67, NCAPD3, NCAPG, NCAPG2, NEK2, NPM1, PCNA, PGAM1, PLK4, PTTG1, RACGAP1, SMC2, SPAG5, STIL, TPX2, UBEC2C, ZWINT*	345	3	EFS 12.7 vs. 26.2 vs. 40.6 months *p* < 0.001	[66]
SKY92/EMC92	92	*SLC30A7, AK2, SYF2, S100A6, NUF2, DARS2, ARPC5, DTL, ANGEL2, LBR, TARBP1, GGPS1,LTBP1, FAM49A, MCM6, ACVR2A, GRB14, ITGA6, DHRS9, STAT1, SPATS2L, BCS1L, SFMBT1, ARL8B, POLQ, MCM2, CCRL1, SEC62, GABRA4, PGM2, NCAPG, FGFR3, SEPT11, AIMP1, CENPE, IL7R 5, DHFR 5, SAR1B, PCDHB7, ATP6V0E1, MCM3, TUBB, TUBB, MARCKS, SLC17A5, NCUBE1, SUN1/GET4, DNAJB9, 208232_x_at, RAB2A, TRAM1, ZNF252, HNRNPK, MRPL41, ZWINT, 243018_at, FANCF, EHBP1L1, C11orf85, PPP2R1B, ROBO3, 238780_s_at, C1S, ESPL1, ITM2B, ZBTB25, NPC2, ATPBD4, C15orf38, FANCI, SMG1, DYNLRB2, TMEM97, SPAG5, TOP2A, BIRC5, C18orf10, TSPAN16, RPS28, RPS11, NOP56, FTL 19, CDH22, DONSON 21, PFKL, ST13, DUX4, RPS4X, KIF4A, HMGN5, HMGB3, MAGEA6*	757 in validation	4	HR 3.4 (95% CI:2.19–5.29, *p* = 5.7 × 10^−8^) in high-risk patients	[68]
CTA	28	*ASPM, BIRC5,* *C4A, CDC2, CENPA, CST3, CTAG2, DLG7, FLJ110719, FLJ21841, GAGE5, GAGED2,* *KIF20A, MAGEA1, MAGEA12, MAGEA3, MAGEA6, NEK2, RACGAP1, RRM2, SGK,* *SSX1, SSX2, SSX3, SSXT/SSX4 fusion, TOP2A, TTK, TYMS.*	53 patients overall	2	Median survival of 27 months in the high-risk cluster 1	[71]
CI	4	*CETN2, TUBG1, PCNT1 PCNT2*	539	2	High CI vs. low CI Median OS 30.6 vs. 45.6 months, log-rank *p* = 0.04 Median PFS 2.8 vs. 4.9 months, log−rank *p* = 0.02	[72]

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
