# Peer review of "The Significance of mRNA in the Biology of Multiple Myeloma and Its Clinical Implications"

_ijms, 2021, doi:10.3390/ijms222112070_

Round 1

Reviewer 1 Report

It is an excellent attempt for a comprehensive review to introduce all methods and achivements  in the field searching for driver mutations and molecular events determinig the course, the prognosis and to herald potential drug resistance in Multiple Myeloma.

This attempt however might not be complete without citing such works like that of Broderick P in Nature Genetics 2011 or Braggio E’s methodology review in  Blood Cancer Journal 2013 or that of the long list of significant contributions of Canzian F and the IMMEnSE consortium to the topic.

Minor comment: in the abbreviation list PCR was written double.

Author Response

 It is an excellent attempt for a comprehensive review to introduce all methods and achivements  in the field searching for driver mutations and molecular events determinig the course, the prognosis and to herald potential drug resistance in Multiple Myeloma.

Response: We thank all the reviewer for the positive review of our paper.

This attempt however might not be complete without citing such works like that of Broderick P in Nature Genetics 2011 or Braggio E’s methodology review in  Blood Cancer Journal 2013 or that of the long list of significant contributions of Canzian F and the IMMEnSE consortium to the topic.

Response: The papers have been discussed and cited – marked yellow.

Minor comment: in the abbreviation list PCR was written double.

Response: Corrected as requested

Reviewer 2 Report

Overall the manuscript is well written, organized, an easy to follow.  Most of the information would be perceived as too basic for an expert in the field, but would be a good introduction to someone new to multiple myeloma genomics.  Importantly, not all previously published gene expression classifiers were mentioned and I feel it would be useful to create a table summarizing the differences between them (e.g. SEMC92, SKY92, UAMS70, UAMS80, UAMS17, IFM15, GEP70, MRCIX6, HM19, GPI50).

Author Response

Overall the manuscript is well written, organized, an easy to follow.  Most of the information would be perceived as too basic for an expert in the field, but would be a good introduction to someone new to multiple myeloma genomics.

Response: We thank all the reviewer for the positive review of our paper

 Importantly, not all previously published gene expression classifiers were mentioned and I feel it would be useful to create a table summarizing the differences between them (e.g. SEMC92, SKY92, UAMS70, UAMS80, UAMS17, IFM15, GEP70, MRCIX6, HM19, GPI50).

Response: The proper table summarizing the differences between genes has been included.

Round 2

Reviewer 2 Report

The authors have added a table which summarizes the various gene expression classifiers for MM which I believe is very useful and elevates the usefulness of this review.